# Pleiotropic Effects of Peroxisome Proliferator-Activated Receptor Alpha and Gamma Agonists on Myocardial Damage: Molecular Mechanisms and Clinical Evidence—A Narrative Review

**DOI:** 10.3390/cells13171488

**Published:** 2024-09-05

**Authors:** María Esther Rubio-Ruíz, Juan Carlos Plata-Corona, Elizabeth Soria-Castro, Julieta Anabell Díaz-Juárez, María Sánchez-Aguilar

**Affiliations:** 1Department of Physiology, Instituto Nacional de Cardiología Ignacio Chávez, Juan Badiano 1, Sección XVI, Tlalpan, México City 14080, Mexico; esther_rubio_ruiz@yahoo.com; 2Department of Interventional Cardiology, Instituto Nacional de Cardiología Ignacio Chávez, Juan Badiano 1, Sección XVI, Tlalpan, México City 14080, Mexico; juancarlosplatacorona3@gmail.com; 3Department of Cardiovascular Biomedicine, Instituto Nacional de Cardiología Ignacio Chávez, Juan Badiano 1, Sección XVI, Tlalpan, México City 14080, Mexico; elizabethsoria824@gmail.com; 4Department of Pharmacology “Dr. Rafael Méndez Martínez”, Instituto Nacional de Cardiología Ignacio Chávez, Juan Badiano 1, Sección XVI, Tlalpan, México City 14080, Mexico; anabelldij@gmail.com

**Keywords:** lipids, cardiovascular diseases, myocardial damage, PPAR agonists, fibrates, thiazolidinediones

## Abstract

Cardiovascular diseases remain the leading cause of death in the world, and that is why finding an effective and multi-functional treatment alternative to combat these diseases has become more important. Fibrates and thiazolidinediones, peroxisome proliferator-activated receptors alpha and gamma are the pharmacological therapies used to treat dyslipidemia and type 2 diabetes, respectively. New mechanisms of action of these drugs have been found, demonstrating their pleiotropic effects, which contribute to preserving the heart by reducing or even preventing myocardial damage. Here, we review the mechanisms underlying the cardioprotective effects of PPAR agonists and regulating morphological and physiological heart alterations (metabolic flexibility, mitochondrial damage, apoptosis, structural remodeling, and inflammation). Moreover, clinical evidence regarding the cardioprotective effect of PPAR agonists is also addressed.

## 1. Introduction

Currently, the treatment of cardio-metabolic diseases is gaining relevance worldwide due to the increase in the incidence of these pathologies. In Mexico, high blood pressure, type 2 diabetes mellitus, and obesity are presented as the non-communicable diseases with the highest percentages of incidence in the population at 15.4%, 12.5%, and 14.8%, respectively [1]. This is why finding an effective and multi-functional treatment alternative to combat these diseases becomes more important. Cardiovascular complications in diabetes mellitus present a burgeoning global public health challenge, significantly amplifying both mortality and morbidity rates among affected individuals [2,3]. A pivotal manifestation of this multifaceted issue is diabetic heart disease, characterized by a notable decline in cardiac contractile function in the absence of ischemia, a condition commonly referred to as diabetic cardiomyopathy.

Among the signaling pathways involved in the pathogenesis of cardiovascular diseases is that of the nuclear peroxisome proliferator-activated receptors (PPARs). PPARs are a group of nuclear regulatory factors that fine-tune key elements of glucose and fat metabolism and regulate inflammatory cell activation and fibrotic processes. There are three isoforms: alpha (α), gamma (γ), and beta/delta (β/δ). Both alpha and gamma isoforms have determined expression patterns. The alpha isoform is mainly expressed in the liver, kidney, muscle, and heart, and the epithelium of blood vessels. In general form, this isoform controls the lipid metabolism. Although PPAR gamma is predominantly expressed in adipose tissue, one of the target tissues for insulin, it has been subsequently found to be expressed in the colon, macrophages, vascular smooth muscle cells, endothelial cells, and several cancer cell lines, as a result of its pleiotropic effects. At the cellular level, it improves insulin sensitivity and glucose homeostasis, and it is involved in adipogenesis. PPAR beta/delta is expressed in all cell types, and this is the reason for its ubiquitous expression in the organism; their functions are sufficiently variable with the expression patterns in the tissue [4]. The natural ligands of both PPAR alpha and PPAR gamma are polyunsaturated fatty acids and eicosanoids derived from cyclooxygenases (COXs) and prostaglandins (PGs). 

PPAR-alpha and PPAR-gamma are nuclear receptors that regulate gene expression related to metabolism and inflammation. In healthy hearts, PPAR-alpha promotes fatty-acid oxidation, providing essential energy, while PPAR-gamma plays a minor role in regulating glucose metabolism and inflammation. In failing hearts, PPAR-alpha is downregulated, leading to reduced fatty-acid metabolism and energy deficiency, contributing to heart dysfunction. PPAR-gamma’s role in heart failure is less clear, showing inconsistent changes that may offer protective anti-inflammatory effects or that could lead to adverse outcomes. Understanding these dynamics is crucial for developing targeted heart-failure therapies [5]. 

PPAR-alpha agonists, such as fibrates, primarily reduce triglyceride levels and may lower cardiovascular disease (CVD) risk, but their effect on heart failure (HF) risk is uncertain. PPAR-gamma agonists, like thiazolidinediones, improve insulin sensitivity and may reduce certain CVD risks, but are associated with increased HF risk due to fluid retention and weight gain. Newer generations of PPAR agonists aim to offer cardiovascular benefits with fewer side effects, although their long-term impacts on HF and CVD require further study [6].

Synthetic PPAR alpha ligands, such as fibrates, are the class of drugs recommended for the control of dyslipidemia, as they reduce triglyceride levels and slightly increase plasma HDL particle levels (5–15%). Moreover, some authors have reported that fibrates such as clofibrate, fenofibrate, and bezafibrate are an effective treatment to reduce myocardial damage in several experimental models by regulating processes such as inflammation, tissue remodeling, and mitochondrial biogenesis. In addition, the activation of PPAR alpha also modulates the activity of the myocardial renin-angiotensin system and restores the insulin-signaling pathway [7].

On the other hand, thiazolidinediones (TZDs), also called glitazones, are a group of oral anti-diabetic drugs designed to treat patients with type 2 diabetes, due to the drugs’ selective binding to PPAR gamma. Rosiglitazone, ciglitazone, troglitazone, and pioglitazone belong to TZD family; these drugs improve insulin sensitivity by acting on adipose, muscle, and liver tissues to increase glucose utilization and decrease glucose production [8].

Recently, new knowledge has been acquired regarding the cardioprotective effects of fibrates and TZDs, beyond their classic effects. Therefore, in the present work we describe the mechanisms underlying the cardioprotective effects of these synthetic PPAR agonists, as well as the clinical evidence.

## 2. Metabolic Effects

Experimental studies have shown that the heart can oxidize different substrates depending on the physiological conditions. The adult heart depends for 40–60% of its energy on the oxidation of fatty acids (FA) (oleic and palmitic, mainly); however, in conditions of low oxygen supply (ischemia), it depends on the oxidation of glucose. This metabolic flexibility confers the advantage of adequately supplying ATP for continual cardiac contraction. Moreover, processes such as transcriptional regulation, mediated via PPARs, contribute to metabolic flexibility by controlling the expressions of proteins that are involved in metabolic pathways [9]. Activation of cardiac PPAR gamma not only increases expressions of genes involved in fatty-acid β-oxidation but also suppresses glucose utilization. TZDs appear to restore insulin sensitivity and enhance insulin action without directly modulating insulin secretion in islet cells [10]. Plasma triglyceride and free fatty-acid levels are markedly reduced in several diabetic rodent models by treatment with TZDs [10]; these drugs also may improve insulin resistance in vivo by normalizing GLUT-4 protein content in adipose tissue and heart, leading to improved glucose metabolism [7,11].

## 3. Local Cardioprotective Effects

### 3.1. Antifibrotic Effect

The fibrotic process is a compensatory mechanism that occurs in order to replace the loss of viable tissue and maintain functionality at the cardiac level. However, this fibrotic scarring can favor the loss of the contractile capacity of the myocytes and it is not reversible; this produces cardiac failure [12]. It has been reported that in this process, inflammatory cells like leukocytes and monocytes (and like macrophages) improve scar formation [13]. 

It has been reported that rosiglitazone and fenofibrate, PPAR gamma and PPAR alpha ligands, are able to prevent vascular hypertrophic growth, as evaluated by the growth index. Also, rosiglitazone improves the endothelial function by acetylcholine reactivity in the DOCA-NaCl 1% hypertension model [14]. Another example is found in Kulkarni’s work, where the 15ΔPGJ2, PPAR gamma endogen ligand inhibits human fibroblast growth by downregulation of TGF-β signaling [15]. This effect is not only at the level of cells of vascular origin. It has even been observed in cells derived from human corneal fibroblasts, where rosiglitazone produces a dose-dependent effect in inhibiting TGF-β and suppressing the formation of extracellular matrix [16]. Likewise, fenofibrate reverses ventricular tissue hypertrophy assessed by reducing the mass of ventricle, collagen deposition, and myocyte diameter and area in a manner dependent on the increase in PPAR alpha expression in the heart [17].

### 3.2. Anti-Apoptotic Effect

It has been showed that in myocardial ischemia, the DNA-binding activity of PPAR/RXR heterodimer is decreased [18]. The importance of PPAR gamma-like cell cycle regulator is evidenced in targeting mutation assays on rats; targeted disruption of PPAR gamma produces embryo lethality due to placental and cardiac defects. It was found that the negative homozygotes do not survive, compared to the wild ones [19]. At heart level, the apoptotic process is present in myocardial ischemia, increasing cell death by oxygen deprivation. Duan et al., showed that PPAR alpha is related to the protective effect against apoptosis and not PPAR gamma, as evaluated by TUNEL assays, hemodynamic parameters, and the expression of proteins related to cell death; however, the crosstalk effect cannot be ignored [20]. Regarding PPAR alpha, it has been reported that fenofibrate decreased aldosterone-induced apoptosis in adult rat ventricular myocytes by blocking the JNK protein and the mediators BAX and caspase 3 [21]. 

### 3.3. Anti-Inflammatory Effect

PPAR gamma is closely related to the inflammatory process. PPAR gamma receptors are highly expressed in monocyte-macrophages and adipose tissue, where they have a role in the inflammatory pathway. An example is found in the study of Fu et al., in which the PPAR gamma expression in macrophages was related to the anti-inflammatory effect in alveolar tissue [22]. 

Its anti-inflammatory properties are well known; the above has been observed in aortic tissue from rodent knockout for the low-density lipoprotein receptor (LDLR −/− mice), which develops atherosclerotic lesions, and in which rosiglitazone, a PPAR gamma ligand (at a dose of 20 mg/kg/day) improved this pathway by increasing scavenger CD36, preventing the lipid deposition on the vessels [23]. Another example is that rosiglitazone produces a reduction of acute inflammation produced by carrageenan 1%, a mucopolysaccharide from red algae which is used to simulate local inflammation. It is mediated by the receptor, as evinced by the observation that PPAR gamma antagonist Bisphenol A diglyccidyl ether attenuated said effect [24,25]. 

Other metabolic pathways are involved in the inflammatory process; one example is cyclooxygenases, and cyclooxygenase 2 (COX-2) is one of them. It was found that 15ΔPGJ2, a PPAR gamma endogen ligand in cultured cardiomyocytes, decreases the expression of COX-2, PGES, and inducible nitric oxide synthase (iNOS) and the production of prostaglandin E2, in a phenomenon dependent on its receptor activation [26]. 

Concerning PPAR alpha, other studies have shown that fenofibrate prevents the progression of hypertensive heart disease by reducing inflammation markers such as nuclear factor kappa B (NF-κB), binding activity and mRNAs of VCAM-1, IL-6, COX-2, and MCP-1, and dysfunction at the myocardial level [27], in addition to reducing myocardial inflammation due to the decreased expression of molecules such as VCAM-1, PECAM, and ICAM-1, in a rat model with angiotensin II (Ang II) infusion [28]. Also, the stimulation of PPAR alpha by fenofibrate has been reported to block the infiltration of T lymphocytes and macrophages in the left ventricle, in addition to decreasing the concentration of C-reactive protein in plasma, which improved survival by attenuating heart failure in a Dahl rat model [29].

On the other hand, diverse studies have shown that the PPAR gamma activation by TDZs and the PPAR alpha activation by fibrates exert an anti-inflammatory and anti-apoptotic effect by the negative modulation of the expression, localization, and release of high mobility group box-1 protein (HMGB1), a nuclear DNA-binding protein which triggers inflammatory processes in response to heart injury [30,31,32].

### 3.4. Effects on Mitochondria

The heart, being a dynamically active metabolic organ abundant in mitochondria, is inherently vulnerable to diminished mitochondrial energetic function [33]. Increased fatty-acid oxidation and mitochondrial activity in the heart can lead to oxidative stress, which may induce pro-inflammatory markers and worsen cardiac function by impairing metabolic activity. The balance between fatty-acid and glucose oxidation, as regulated by PPARs, is crucial for maintaining cardiac health. Disruptions in this balance, such as those seen in heart failure, can lead to adverse outcomes involving both oxidative stress and inflammation [34].

Mitochondria are subject to several processes which occur continuously and are part of the phenomenon known as “mitochondrial dynamics”. These processes comprise the formation of new mitochondria (biogenesis) and their elimination (mitophagy), as well as fusion/fission and intracellular transport. Mitochondrial fusion occurs when two organelles join and form a larger one and can even form networks; the opposite process is known as fission, in which the organelle fragments into two units [35]. It is important to highlight several aspects of mitochondrial dynamics: (1) they depend on the cell type and its metabolic state; (2) fusion and fission processes must be maintained in balance to ensure mitochondrial stability and functionality; (3) the processes are controlled by groups of independent proteins, the activity of which can be regulated by post-translational modifications; and (4) they are subject to modifications derived from some physio-pathological stimuli. Numerous studies have shown that heart diseases are associated with alterations in mitochondrial dynamic processes, as well as with the dysfunction of this organelle [35]. 

The modifications in dynamic mitochondrial cause an increase in the production of reactive oxygen species (ROS) and a decrease in oxidative phosphorylation, which ultimately promotes the cell death of cardiomyocytes. In the context of diabetes, an increase in mitochondrial fission and decreased peroxisome-proliferator-activated receptor gamma coactivator 1-alpha (PGC1α) expression, the major regulator of mitochondrial biogenesis are observed in cardiac muscle cells [36,37,38]. The inhibition of mitochondrial fission emerges as a promising therapeutic target, warranting thorough investigation to unravel its potential in ameliorating the impacts of diabetes on mitochondrial and cardiac function. In this context, experimental evidence has shown that fibrates can preserve mitochondrial function by increasing PGC1α expression and thus improve the ability to produce ATP [39,40]. It has been reported that TZDs upregulate mitochondrial oxidative phosphorylation, mitochondrial biogenesis, and antioxidant defense. Moreover, TDZs can decrease mitochondrial damage by increasing the expression of Small Heat Shock Protein 22 (HSP22) a key protein involved in cardiac protection under various conditions of myocardial ischemic stress in animal models and patients. On the other hand, the PPAR gamma activation by rosiglitazone promotes mitochondrial biogenesis and reduces ROS formation [41].

Experimental studies have revealed an elevation in mitochondrial calcium uniporter beta subunit (MCUb) levels in type 2 diabetes models, which is attributed to reduced nuclear receptor corepressor 2 (Ncor2) levels, diminished transcriptional repression of Ncor2, and heightened transcriptional activity of PPAR alpha. In muscle, Ncor2 acts as a repressor for genes crucial in fatty-acid catabolism, forming associations with nuclear receptors like PPARs in the absence of ligands to suppress transcriptional activity [42]. The downregulation of Ncor2, coupled with the activation of PPAR alpha, constitutes a mechanism through which the type 2 diabetic heart adapts metabolically to glucose intolerance and hyperlipidemia. This adaptation involves the upregulation of MCUb, ultimately restricting Ca^2+^ influx in the mitochondrial matrix and favoring FA over glucose for energy production [42]. 

### 3.5. Effects on Sirtuins

Sirtuins (SIRT1-SIRT7) are a family of NAD^+^-dependent deacetylases and ADP-ribosyl transferases which target histones and non-histone proteins that are involved in processes such as cell proliferation, cell differentiation, DNA damage repair, energy homeostasis, metabolic and oxidative stress resistance, and tissue regeneration and inflammation; this is why these proteins have emerged as a target in the prevention and treatment of CVD [43]. Sirtuins have different subcellular locations, depending on the different kinds of cell types and specific circumstances: SIRT1 and 2 are mainly localized in the nucleus and cytosol, and SIRT3–5 are located in mitochondria; meanwhile, SIRT6 and SIRT7 are nuclear proteins. 

Studies in several models, such as cardiac hypertrophy and ischemia/reperfusion injury, have shown that SIRT1 has a cardioprotective effects through PGC-1α-dependent mechanisms by decreasing oxidative stress, ET-1 production, and pro-inflammation factors; inhibiting apoptosis; modulating eNOs activity; and promoting autophagia [44,45]. SIRT3 increases energy production by activating the mitochondrial electron transport chain, upregulating fatty-acid oxidation and is also involved in the defense against oxidative stress. Moreover, several studies have also shown an anti-inflammatory effect of SIRT1–3 and 6 by its regulation of inflammatory pathway components such as NF-κB, toll like receptor 2 (TLR2), and pyrin domain-containing protein 3 (NLRP3) inflammasome [46]. The evidence supports the determination that the activation of SIRT6/PPARα pathway regulates the expression of fatty-acid oxidation-related genes, induces mitochondrial biosynthesis, ameliorates oxidative stress, decreases cardiac fibrosis, and reduces infarct size, improving the heart function. In addition, the cardioprotective effects of SIRT6 involve the activation of other signaling factors such as FOXO1, FOXO3, AMP-activated protein kinase (AMPK), and Akt which might act synergistically [47].

It has been reported that fenofibrate treatment prevents myocardial damage by increasing FGF21 expression, a cardiac chemokine that regulates energy balance and maintains the cellular homeostasis that increases SIRT1-dependent autophagy [48].

### 3.6. Effects on Vasoactive Agents

The effects on the production of various peptides through the stimulation of PPARs have been widely studied. In particular, endothelin (ET-1) is a mitogen factor that is involved in myocardial hypertrophy. Also, ET-1 is related to increased cell-size in human cardiomyocytes in a dependent manner with adiponectin and NF-κB factor; this effect was reversed by the action of fenofibrate. Fenofibrate treatment prevents cardiac hypertrophy by decreasing the expression of ET-1 mRNA, as well as collagen type I and type III mRNA in a pressure-overload model in rats by abdominal aorta banding [49]. This beneficial effect of PPAR alpha stimulation was corroborated using silencing of PPAR alpha with siRNAs [50,51]. On the other hand, Iglarz et al. found that PPAR gamma and alpha stimulation decreased the mRNA of ET-1 and improved vascular reactivity in nephrectomy plus DOCA salt rats [14]. 

In addition to the profibrotic role of ET-1, Ang II also favors this effect. Diep’s group has reported that fenofibrate prevents myocardial inflammation and collagen deposition resulting from perfusion with Ang II in a murine model [28]. Our group has reported that in the metabolic syndrome model (30% sucrose intake), the fenofibrate therapy significantly reduced Ang II concentration and downregulated the Ang II signaling pathway, in addition to decreasing myocardial fibrosis and restoring local insulin sensitivity [7,52].

Nitric oxide (NO) has been described as a potent vasodilator, but it also plays an important anti-atherogenic role by inhibiting smooth-muscle proliferation in vascular tissue, platelet accumulation, and endothelial cell–leukocyte interaction. NO is enzymatically produced by three isoforms of NO synthase (NOS), namely, nNOS or NOS1; inducible NOS, iNOS, or NOS2; and eNOS or NOS3, expressed in neuronal, immune, and endothelial cells, respectively [53]. Experimental and clinical forms of evidence have shown that PPARα activation by fibrates regulates the expression of cardiac eNOS and increases the phosphorylation of eNOS in serine 1177 residue (an eNOS activation site), which increases NO production. Moreover, fibrates also increase BH_4_ concentration, a cofactor necessary for eNOS coupling, and decrease oxidative stress, which in turn, increase the production and bioavailability of NO [52,54]. Additionally, fibrates have an anti-inflammatory effect, which is achieved by decreasing the expression and activity of iNOS [55].

On the other hand, due to eNOS activity being critically regulated by insulin, TDZs increase NO production by enhancing insulin sensitivity and diminishing oxidative stress [56,57].

## 4. Clinical Evidence

### 4.1. Fibrates

Clinically, the activation of PPARs stands as a significant strategy used to address insulin resistance and dyslipidemia, with PPAR gamma activation by TDZs and PPAR alpha activation by fibrates playing distinct roles [58].

Fibrates are agonists of PPAR alpha, acting via transcription factors regulating, among other things, various steps in lipid and lipoprotein metabolism. Therefore, fibrates have good efficacy in lowering fasting triglyceride levels, as well as post-prandial TG and TG-rich lipoprotein remnant particles [59]. Clinical impacts on lipid profiles vary among members of the fibrate class, and with the patient’s pretreatment lipoprotein status, as well as the relative potency of the fibrate used, but are estimated to reach a 50% reduction of the TG level, a <20% reduction of the low-density lipoprotein cholesterol (LDL-C) level, and an increase of the high-density lipoprotein cholesterol (HDL-C) level of <20% [60].

Elevated plasma triglyceride levels are associated with an increasing risk of atherosclerotic cardiovascular disease (ASCVD); this association becomes null after adjusting for non-HDL-C, an estimate of the total concentration of all ApoB-containing lipoproteins [61]. The clinical effects of fibrates have been primarily illustrated by six randomized controlled trials (RCT) [62,63,64,65,66] (Table 1). In the findings from those trials, the lipid profile benefit is obvious, but cardiovascular outcomes showed different results. Given this information, there are just a few conditions in which fibrates are indicated; we present the classic general indication for fibrates and then the specific clinical indications. The classic indication for fibrate treatment in the general population with hypertriglyceridemia, in accord with recent guidelines, is the following:

In primary prevention, for patients who are at LDL-C goal with triglyceride levels >2.3 mmol/L (>200 mg/dL), fenofibrate or bezafibrate may be considered in combination with statins (IIb B).

In high-risk patients who are at LDL-C goal with triglyceride levels >2.3 mmol/L (>200 mg/dL), fenofibrate or bezafibrate may be considered in combination with statins (IIb C) [59].

**Table 1 cells-13-01488-t001:** Classical clinical effects of fibrates.

	Helsinki Heart Study (HHS) [61]	Veterans Affairs High Density Lipoprotein Intervention Trial (VA-HIT) [62]	The Bezafibrate Infarction Prevention (BIP) Study [63]	Lower Extremity Arterial Disease Event Reduction (LEADER) [64]	Fenofibrate Intervention and Event Lowering in Diabetes (FIELD) [65]	The ACCORD-Lipid Study: Implications for Treatment of Dyslipidemia in Type 2 Diabetes Mellitus [66]
Year	1987	1999	2000	2002	2005	2011
Drug	Gemfibrozil	Gemfibrozil	Bezafibrate	Bezafibrate	Fenofibrate	Simvastatin + Fenofibrate
Dosage	600 mg TD	1200 mg OD	400 mg OD	400 mg OD	200 mg OD	Simvastatin 20–40 mg OD and fenofibrate 160 mg OD
Type of prevention	Primary	Secondary	Secondary	Secondary	Primary	Primary
Intervention	Randomized, double-blind drug vs. placebo	Drug vs. placebo	Randomized, double- blinded drug vs. placebo	Randomized, blinded drug vs. placebo	Double-blind, placebo-controlled trial	Randomized, multicenter, double 2 × 2 factorial design study
Mean follow-up	5 years	5 years	6.2 years	4.6 years	5 years	4.7 years
Patients enrolled	4081	2531	3090	1568	9795	5518
Inclusion criteria or lipid cut-off	Healthy Patients with non-HDL-C greater than or equal to 200 mg per deciliter	Male < 74 years old with document CAD and the following lipid profile: HDL ≤ 40, LDL ≤ 140, TG ≤ 300.	Patients 45–74 years of age with a history of MI and/or angina TC 180–250 mg/dL, TG < 300 mg/dL, LDL < 180 mg/dL, and HDL < 45 mg/dL	Men with lower-extremity arterial disease.	Patients with T2D and TG 2.31 mmol/L (≥204 mg/dL)	T2D patients who are at high risk for CVD events because of existing CVD or additional risk factors. TG 2.31 mmol/L (≥204 mg/dL)
Effects in lipid profile	>10% HDL, TC < 11%, <LDL-C 10%, <TG 43%	>6% HDL-C, <31% TG, <4% TC. No differences in LDL	<TG 21%, >HDL 18%	<7.6% TC, <8.1% LDL-C, >8% HDL-C, <23% TG	<11% TC, <12% LDL-C, <29% TG, >5% HDL-C	>HDL 6.3%, <21% TG, No difference in LDL-C between groups
Clinical benefits	− <37% non-fatal AMI and 34% in CAD − No differences in total death (TD) between groups.	<22% of CHD death or AMI. <24% in the composite endpoint of CHD death, nonfatal AMI and stroke.	− No differences in fatal AMI, non-fatal AMI or morality. − Reduction in the cumulative probability of the primary endpoint by Bezafibrate was 39.5% in subgroup with high baseline triglycerides (>200 mg/dL).	− No differences in fatal AMI, non-fatal AMI or mortality.	− No differences in myocardial infarction or CHD death. − 24% reduction in non-fatal AMI, together with a significant 21% relative reduction in coronary revascularization. − Less albuminuria progression and retinopathy needing laser treatment.	− Combination therapy did not significantly reduce (coronary heart disease and stroke) event rates. − Subgroup with hypertriglyceridemia and low HDL-C experienced a 31% lower event rate with combination therapy.

#### Indications and Clinical Use of Fibrates in Specific Lipoprotein Disorders

Fibrates are first-line drugs for the treatment of primary hypertriglyceridemia. In these patients, fibrates most noticeably decrease plasma TRLs; they also decrease, albeit to a lesser extent, total cholesterol, whereas HDL-C levels increase [67].

Type III Dysbetalipoproteinemia is a rare lipid disorder resulting from homozygosity for the rare apoE2 isoform in predisposed subjects. The characteristic disturbance of this metabolic disorder is the accumulation of cholesterol-enriched VLDL, which migrates in β-position on agarose gel electrophoresis. Fibrates have a spectacular lipid-lowering potential in patients with this disorder. The levels of circulating triglyceride and cholesterol are greatly diminished. Simultaneously, LDL-C and HDL-C, which are usually low, increase significantly [68].

As for primary hypercholesterolemia, although fibrates are not considered to be first-line drugs, the new generation of fibrates efficiently reduces plasma cholesterol and LDL-C and increases HDL-C concentrations when used in monotherapy in patients with primary hypercholesterolemia [69].

There is accumulating evidence that fibrates positively impact microvascular disease in patients with type 2 diabetes. In both FIELD [65] and ACCORD-Lipid [66], fenofibrate reduced the development and progression of albuminuria, indicating a positive impact on the progression of diabetic renal disease. 

In addition, in both studies, fenofibrate treatment reduced the progression of diabetic retinopathy, as reflected by the progression of retinopathy on fundoscopic examination and the need for laser photocoagulation. In a particularly intriguing post hoc analysis, the FIELD investigators observed that fenofibrate use resulted in a significant reduction in the risk of lower-extremity amputation (hazard ratio: 0.64; 95% CI: 0.44–0.94), with primarily minor amputation in participants without known large-vessel disease [70]. Thus, independent of considerations regarding CVD prevention, the presence of diabetic nephropathy, retinopathy, and/or disordered lower extremity microcirculation may warrant consideration of fibrate therapy in patients with type 2 diabetes, because fenofibrate treatment may produce beneficial effects over diabetic macular edema and retinopathy for inflammation. 

There is evidence that fibrates have an advantageous influence on inflammatory and thrombogenic plasma risk factors in patients with dyslipidemia [71]. Other studies, such as Ghani et al., have focused on type 2 diabetes and the early development of endothelial dysfunction. In this study, the effects of fenofibrate on inflammatory markers, metabolic parameters, and endothelial dysfunction were determined and it was concluded that fibrates have improvements in endothelial function, independent of the lipid-level impact [72]. In summary, although the overall impact of fibrate therapy on CVD in large clinical trials has varied, most likely owing to differences in the populations studied, within each of these trials there is a clear and consistent finding of CVD risk reduction with fibrate therapy in a subgroup of participants characterized by the presence of significant hypertriglyceridemia and/or low HDL-C. Overall benefit of improvement in lipid profile using fibrates does not necessary translate into better clinical cardiovascular outcomes.

### 4.2. Thiazolidinediones or Glitazones

Thiazolidinediones and glitazones (TZDs) are potent insulin sensitizers and significantly improve glycemic control. Though they are not first-line agents, both rosiglitazone and pioglitazone are currently used in the treatment of type 2 diabetes mellitus, alone or in combination with sulfonylurea, metformin, or insulin [73]. 

It has been reported that TZDs, by their binding to PPAR gamma, modulate the transcription of genes of carbohydrates and lipid metabolism, improving insulin sensitivity, dyslipidemia, adipose tissue remodeling, and adiponectin synthesis [74,75]. Apart from their function in glycemic control and improvement of insulin resistance, TZDs have anti-inflammatory and antioxidant properties that provide them with cardiovascular benefits. Clinical and experimental observations suggest that TZDs have anti-atherosclerotic effects by increased nitric oxide bioavailability, decreased leukocyte/endothelial cell interaction, inhibition of the production of inflammatory cytokines, reduced vascular smooth-muscle cell migration and proliferation, and cholesterol efflux from macrophages [76,77].

Vein-Coronary Atherosclerosis and Rosiglitazone after Bypass Surgery (VICTORY) was the first cardiometabolic study that evaluated the prevention of atherosclerosis progression in patients with type 2 diabetes, but the results were inconclusive [78]. In, 2010, the Assessment on the Prevention of Progression by Rosiglitazone on Atherosclerosis in Diabetes Patients with Cardiovascular History (APPROACH) study showed that rosiglitazone did not significantly decrease the primary endpoint of progression of coronary atherosclerosis in patients with type 2 diabetes mellitus [79]. Zhou et al., in 2017, demonstrated that the treatment with TZDs is associated with a significant reduction in in-stent restenosis, target lesion revascularization, and MACE in patients after percutaneous coronary intervention [80] (Table 2).

However, there is still currently controversy regarding the association between TZDs and myocardial infarction. Some clinical studies suggested that rosiglitazone was associated with an increased cardiovascular risk [81]; however, the reevaluation of the Rosiglitazone Evaluated for CV Outcomes and Regulation of Glycemia in Diabetes (RECORD) trial confirmed no increased risk for heart attack in individuals with rosiglitazone treatment versus diabetes medications [82]. On the other hand, the Prospective Pioglitazone Clinical Trial in Macrovascular Events (PROactive), demonstrated that pioglitazone has a protective effect for cardiovascular disease, including heart failure events in patients with diabetes; the study shows that three outcomes of the main secondary endpoint were improved [83]. 

Despite the side effects described to TDZ, there are new data that show the positive effects of these drugs. Xue et al. demonstrated that TDZs play a protective role in myocardial infarction by raising adiponectin, protecting the endothelium, and delaying plaque development, independently of the status of serum lipids [75]. Other clinical studies showed that pioglitazone treatment is associated with significantly lower risks of death, myocardial infarction, and stroke among a diverse population of patients with diabetes [84,85]. There is information that supports the pleiotropic benefits of TZDs in reducing major cardiovascular events and mortality in DT2 and non-DT2 patients, especially in those patients with a high risk of macrovascular events, and this is non-related with improvement in lipid profile. Table 2 summarizes some clinical trials of cardioprotective effects of TDZs.

**Table 2 cells-13-01488-t002:** Clinical effects of thiazolidinediones.

	Effects of Thiazolidinediones on In-Stent Restenosis and Target Lesion Revascularization: A Meta-Analysis of Randomized Controlled Trials [80]	Secondary Prevention of Macrovascular Events in Patients with Type 2 Diabetes in the PROactive Study (PROspective Pioglitazone Clinical Trial in Macrovascular Events): A Randomized Controlled Trial [83]	Thiazolidinediones Play a Positive Role in the Vascular Endothelium and Inhibit Plaque Progression in Diabetic Patients with Coronary Atherosclerosis: A Systematic Review and Meta-Analysis [75]	Pioglitazone and Risk of Cardiovascular Events in Patients with Type 2 Diabetes Mellitus: A Meta-Analysis of Randomized Trials [84]	Pioglitazone after Ischemic Stroke or Transient Ischemic Attack [85]
Year	2017	2005	2022	2007	2016
Drug	Rosiglitazone and Pioglitazone	Pioglitazone	Rosiglitazone and Pioglitazone	Pioglitazone	Pioglitazone
Dosage	Pioglitazone 30 mg/OD, Rosiglitazone 4 mg/OD	15–45 mg, depending on tolerability	Rosiglitazone 4 mg/OD, Pioglitazone 15–30 mg/OD	15–30 mg/OD	45 mg/OD
Type of prevention	Secondary	Secondary	Secondary	Primary	Primary
Intervention	Drug vs. placebo or other anti-diabetic therapy	Drug vs. placebo	Drug vs. placebo	Drug vs. placebo or active comparator	Drug vs. placebo
Mean follow-up	6–18 months	34.5 months	3–12 months	3–24 months	4.8 years
Patients enrolled	1350	5238	451	16 390	3876
Inclusion criteria	Post-percutaneous intervention patients	Patients with T2DM + >6 months macrovascular event	Patients with T2DM + and coronary heart disease or vascular stent surgery suggesting coronary atherosclerosis	Adult patients with T2DM and inadequate glycemic control	Patients with recent ischemic stroke or transient ischemic attack, non-T2DM patients
Endpoints	The number of patients with angiographic in-stent restenosis and patients required to have target lesion revascularization during follow-up.	Primary: Composite of all-cause mortality, non-fatal infarction, stroke, cute coronary syndrome, intervention in leg or amputation. Secondary endpoint: composite of all-cause mortality, non-fatal myocardial infarction, and stroke.	Changes in vascular endothelial and plaque-related indices after treatment in patients with diabetes combined with coronary atherosclerosis, and to explore potential targets for the protective effects of TZDs in myocardial infarction.	Composite of death, myocardial infarction, or stroke. Secondary outcome measures included the incidence of serious heart failure.	Fatal or nonfatal stroke or myocardial infarction.
Outcomes	Reduction in ISR, target lesion revascularization and major adverse cardiac events.	No significant reduction of the primary endpoint. Secondary endpoint reduced with statical significance. (0.84, 0.72–0.98, *p* = 0.027).	Inhibitory effect on plaque progression and a protective effect on the vascular endothelium in patients with diabetes and coronary atherosclerosis.	Lower risk of death, myocardial infarction, or stroke among a diverse population of patients with diabetes. Risk of serious heart failure is increased by pioglitazone, although without an associated increase in mortality.	The risk of stroke or myocardial infarction was lower among patients who received pioglitazone than among those who received placebo. Pioglitazone was also associated with a lower risk of diabetes.

## 5. Limitations

The present work represents a descriptive overview regarding the local cardioprotective effects of synthetic PPAR agonists as well as the clinical evidence to support our point of view; as a result, the search criteria may reflect some bias.

## 6. Conclusions and Future Perspectives

Fibrates and thiazolidinediones, PPAR alpha and gamma activators, are the pharmacological therapies used to treat dyslipidemia and type 2 diabetes, respectively. New mechanisms of action of these drugs have been found, demonstrating their potent cardioprotective effect at different cellular levels beyond their classical action. Randomized controlled studies should be conducted to expand the current indications, which now are mainly focused on biochemical control. The study of the mechanisms of action of currently available drugs, combined with the development of new therapeutic agents, will provide promising options for the treatment of cardiac pathologies.

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
