# Peer review of "Pleiotropic Effects of Peroxisome Proliferator-Activated Receptor Alpha and Gamma Agonists on Myocardial Damage: Molecular Mechanisms and Clinical Evidence—A Narrative Review"

_cells, 2024, doi:10.3390/cells13171488_

Round 1

Reviewer 1 Report

Comments and Suggestions for Authors

In this review article, the authors present the results of studies on possible beneficial effects of PPAR agonists, prescribed for treatment of hypertension, high cholesterol, or type II diabetes, on the heart in a cardioprotective manner. The review is thorough and well balanced, including inconclusive studies along with ones showing positive effects. I have the following minor comments:

1) Section 3: The phrase should be "fibrotic scar" not "fibrotic scare"

2) The authors should have a section explaining the specific targets, actions, and differences in the effects of the various fibrates and TZDs referenced throughout the paper. The many forms of each of these drug families gets confusing.

3) The authors should define the herbal treatments or irritants mentioned through the review, such as carrageenan.

4) The fibrates guidelines would be better presented in a table.

Comments on the Quality of English Language

This paper needs extensive English language editing for grammar, especially rewriting many sentence fragments, and for typos (ex. "fibrotic scare" instead of "scar")

Author Response

Reviewer 1

In this review article, the authors present the results of studies on possible beneficial effects of PPAR agonists, prescribed for treatment of hypertension, high cholesterol, or type II diabetes, on the heart in a cardioprotective manner. The review is thorough and well balanced, including inconclusive studies along with ones showing positive effects. I have the following minor comments:

  1. Section 3: The phrase should be "fibrotic scar" not "fibrotic scare"

R= Thank you for your observation. We corrected the sentence.

  1. The authors should have a section explaining the specific targets, actions, and differences in the effects of the various fibrates and TZDs referenced throughout the paper. The many forms of each of these drug families gets confusing.

R= Thank you for your suggestion. We have included more information regarding fibrates and TZDs to clarify (lines 75-91).

  1. The authors should define the herbal treatments or irritants mentioned through the review, such as carrageenan.

R= Thank you for your suggestion. We have included the information regarding carrageenan and we have included the reference (lines 149-152). Reference: Fehrenbacher, J. C. & McCarson, K. E. Models of inflammation: carrageenan air pouch. Current Protocols. 2021 1, e183. doi: 10.1002/cpz1.183

  1. The fibrates guidelines would be better presented in a table.

R= There is just two formal indications for fibrates in recent guidelines. That’s why we only consider mentioning them.

  1. This paper needs extensive English language editing for grammar, especially rewriting many sentence fragments, and for typos (ex. "fibrotic scare" instead of "scar")

R= We corrected the sentence, thank you.

Reviewer 2 Report

Comments and Suggestions for Authors

Review for the manuscript

Pleiotropic effects of PPAR alpha and gamma agonists on myocardial damage. Molecular mechanisms and clinical evidence

Dear Editor,

Thank you for the opportunity to review this manuscript for Cells. I have some comments and suggestions before it can be accepted for publication in this Journal.

OVERALL COMMENTS

            Based on the knowledge that Fibrates and thiazolidinediones, PPAR alpha, and gamma activators are important therapies to treat dyslipidemia and type 2 diabetes, new mechanisms of action of these drugs have been elucidated, showing their effects to preserve the heart and myocardial damage. From there, the authors performed a review that compared the cardioprotective effects of PPAR agonists by regulating morphological and physiological heart alterations.

TITLE

            In the title, I suggest that the authors define PPAR since they put PPAR alpha and gamma or simply use PPARα and γ. Moreover, clearly show that this is a review and what type of review it is.

ABSTRACT

            The Abstract is fine. However, I suggest that the authors check MDPI instructions. A structured abstract is recommended now.

Please define PPAR.

KEYWORDS

The authors have chosen ": myocardial damage; PPAR agonists; fibrates; thiazolidinediones; pleiotropic effects" as key-words. I suggest: lipids, cardiovascular diseases, myocardial damage; PPAR agonists; fibrates; thiazolidinediones

INTRODUCTION

I understand and agree with the references included in the Introduction. I also know that this is a short review. However, since it is a review, I strongly suggest that the authors expand this section and include references published in the last two years. It is a review and there is only one citation from 2024 and only 3-5 from 2023. I suggest that they keep the article well updated so that anyone who uses this study as a basis for others or as clinical conduct has the most recent information possible.

The aim of this study is clear in the Abstract. However, it is not clear after the Introduction section. Please include a clear objective that justifies the building of this review. Moreover, is this a narrative review?

Why did the authors not performed a systematic review using this interesting topic?

Line 63:

Please expand Metabolic effects. There is more to explore in this section. With the lipid oxidation and "hard work" of mitochondria, is there a rise in oxidative stress that could contribute to heart damage? Can pro-inflammatory markers be produced so that this scenario (free radicals and inflammation) could worsen metabolic activity? Do the included drugs affect these pathways?
            I see the answers for many of these questions in section 3 and its sub-sections. However, I insist that an overview of my mentioned concerns should be included in section 2. The local effects can be maintained in section 3 as they are.

CLINICAL EVIDENCE

Although Table 1 contains very old studies (didn't the authors find any newer studies?), it is clear.

I do not consider it appropriate to place a conclusion at the end of Table 1. This should be done in the text and not as part of the table.

Table 2 is very confusing. Perhaps the authors should reduce the information a little so that it is more accessible to the lay reader.

As above, do not include a conclusion at the end of Table 2. This should be done in the text and not as part of the table.

DISCUSSION

This section needs the inclusion of newer references (as I pointed-out in the Introduction).

CONCLUSION

This section is adequate.

I suggest including an item talking about the strengths and limitations of this review.

I also suggest including a "Future perspectives" item. The review will be much better.

Comments on the Quality of English Language

Minor

Author Response

TITLE

  1. In the title, I suggest that the authors define PPAR since they put PPAR alpha and gamma or simply use PPARα and γ. Moreover, clearly show that this is a review and what type of review it is.

R=Thank you for your suggestion. We added the PPAR definition, and we added the sentence: narrative review in the title, to specify.

 ABSTRACT

  1. The Abstract is fine. However, I suggest that the authors check MDPI instructions. A structured abstract is recommended now. Please define PPAR.

R=Thank you for your suggestion. We added the PPAR definition, and we followed the MDPI recommendations.

  1. KEYWORDS

The authors have chosen ": myocardial damage; PPAR agonists; fibrates; thiazolidinediones; pleiotropic effects" as key-words. I suggest: lipids, cardiovascular diseases, myocardial damage; PPAR agonists; fibrates; thiazolidinediones

R= We changed the keywords according to reviewer suggestion, thank you.

 INTRODUCTION

  1. I understand and agree with the references included in the Introduction. I also know that this is a short review. However, since it is a review, I strongly suggest that the authors expand this section and include references published in the last two years. It is a review and there is only one citation from 2024 and only 3-5 from 2023. I suggest that they keep the article well updated so that anyone who uses this study as a basis for others or as clinical conduct has the most recent information possible.

R= Thank you for your observation. We update the references in our manuscript with newer studies that cover the same or similar topic as far as possible. However, we could not replace the references of classic studies with significant evidence.

  1. The aim of this study is clear in the Abstract. However, it is not clear after the Introduction section. Please include a clear objective that justifies the building of this review. Moreover, is this a narrative review? Why did the authors not performed a systematic review using this interesting topic?

R= We added the aim in introduction section (lines 89-91) according to reviewer suggestion.

In the present work (a narrative review), we synthesize information into a reader-friendly format with a broader scope, on the cardioprotective effects of synthetic PPAR alpha and gamma agonists beyond its pharmacological classical effects. In contrast, systematic reviews answer a narrow question through detailed and comprehensive literature searches. However, at the present, we do not have the quantitative methods to carry out this type of analysis and analyze all the available evidence in scientific databases.

  1. Line 63: Please expand Metabolic effects. There is more to explore in this section. With the lipid oxidation and "hard work" of mitochondria, is there a rise in oxidative stress that could contribute to heart damage? Can pro-inflammatory markers be produced so that this scenario (free radicals and inflammation) could worsen metabolic activity? Do the included drugs affect these pathways? I see the answers for many of these questions in section 3 and its sub-sections. However, I insist that an overview of my mentioned concerns should be included in section 2. The local effects can be maintained in section 3 as they are.

 R= The reviewer was right in her/his observation. Increased fatty acid oxidation and mitochondrial activity in the heart can lead to oxidative stress, which may induce pro-inflammatory markers and worsen cardiac function by impairing metabolic activity. The balance between fatty acid and glucose oxidation, regulated by PPARs, is crucial for maintaining cardiac health. Disruptions in this balance, such as those seen in heart failure, can lead to adverse outcomes involving both oxidative stress and inflammation. We have added a general information and reference (lines 175-181).

CLINICAL EVIDENCE

  1. Although Table 1 contains very old studies (didn't the authors find any newer studies?), it is clear.

R= The current guidelines are based on classic studies with significant evidence, those studies are cited in our manuscript, it is worth mentioning that after an exhaustive information search, there is not much current information on the pleiotropic effects of these drugs.

  1. I do not consider it appropriate to place a conclusion at the end of Table 1. This should be done in the text and not as part of the table.

R= Thank you for your observation. We deleted the sentence of the table and we have included in the text (lines 350-356).

  1. Table 2 is very confusing. Perhaps the authors should reduce the information a little so that it is more accessible to the lay reader. As above, do not include a conclusion at the end of Table 2. This should be done in the text and not as part of the table.

R= Thank you for your observation. We reduced the information of the table as far as possible and deleted the conclusion and we have included in the text (lines 398-401).

 DISCUSSION

  1. This section needs the inclusion of newer references (as I pointed-out in the Introduction).

R= Thank you for your observation. We update the references in our manuscript with newer studies that cover the same or similar topic as far as possible. However, we could not replace the references of classic studies with significant evidence.

CONCLUSION

This section is adequate.

  1. I suggest including an item talking about the strengths and limitations of this review. I also suggest including a "Future perspectives" item. The review will be much better.

R= Thank you for your suggestion. We added the information in sections 5 and 6 of our manuscript.

Reviewer 3 Report

Comments and Suggestions for Authors

The authors submitted the narrative review in which they elucidated cardiac protective mechansms and clinical evidence of it among PPAR alpha and gamma activators. ALthough the authors gave in general positive characteristics of the agents in connection with dyslipidemia management, I would like to recommend the following:

1. There is no clear explanation of SIRT6-depending mechanisms of myocardial protection from damage including following chemotherapy with the PPAR agonists. Please, extend the section of the underlying molecular mechanisms accordingly.

2. Please, consider PPAR alpha signalling pathway, but not PPAR gamma, as a mechanism against myocardial reperfusion damage and add more information regarding it. Include please the explanation of the role of  PPAR alpha/gamma - SIRT1-PGC1α axis

3. Please, report clear explanation regarding inhibiting inflammation and apoptosis via the PPARγ/HMGB1/NLRP3 axis

4. Dispute please PPAR-alpha and -gamma expression and activity in failing heart and in physiological condition

5. Elucidate please the impact of PPAR alpha / gamma agonists on CVD risk and HF risk depending on generation of these agents.

Author Response

  1. There is no clear explanation of SIRT6-depending mechanisms of myocardial protection from damage including following chemotherapy with the PPAR agonists. Please, extend the section of the underlying molecular mechanisms accordingly.

R= Thank you for your suggestion. We have included information regarding the effect of PPAR agonists on sirtuins signaling pathway and the corresponding references in the section 3.5 of the manuscript (lines 223-249).

  1. Please, consider PPAR alpha signalling pathway, but not PPAR gamma, as a mechanism against myocardial reperfusion damage and add more information regarding it. Include please the explanation of the role of PPAR alpha/gamma - SIRT1-PGC1α axis

R= Thank you for your suggestion. We have included information regarding the effect of PPAR agonists on PPAR alpha/gamma - SIRT1-PGC1α axis and the corresponding references in the point 3.5 of the manuscript.

  1. Please, report clear explanation regarding inhibiting inflammation and apoptosis via the PPARγ/HMGB1/NLRP3 axis

R= We have included information regarding the effect of PPAR agonists on the PPARγ/HMGB1/NLRP3 axis in cardiac disease according to reviewer suggestion (lines 168-172).

  1. Dispute please PPAR-alpha and -gamma expression and activity in failing heart and in physiological condition

R= We added the information in introduction section (lines 59-67) according to reviewer suggestion and we have included the reference. Thank you.

  1. Elucidate please the impact of PPAR alpha / gamma agonists on CVD risk and HF risk depending on generation of these agents.

R= We added the information in introduction section (lines 68-74) according to reviewer suggestion and we have included the reference. Thank you.

Round 2

Reviewer 2 Report

Comments and Suggestions for Authors

Dear authors,
Thank you for performing the modifications.
Your manuscript can be accepted.

Comments on the Quality of English Language

Minor.

Reviewer 3 Report

Comments and Suggestions for Authors

The authors submitted the critically revised version of the paper along with the concose reply to the revierwers. I am satisfied with the corrections and have no serious concerns about the manuscript in its revised version.